# Outcomes and clinical implications of intranasal insulin on cognition in humans: A systematic review and meta-analysis

Sally Wu[1,2‡], Nicolette Stogios[1,2‡], Margaret Hahn[1,2,3,4], Janani Navagnanavel[5], Zahra Emami[6], Araba Chintoh[1,2,3], Philip Gerretsen[2,3,7], Ariel Graff-Guerrero[2,3,7], Tarek K. Rajji[2,3,8], Gary Remington[1,2,3], Sri Mahavir Agarwal[1,2,3,4]*

1 Schizophrenia Division, Centre for Addiction and Mental Health (CAMH), Toronto, Canada, 2 Institute of Medical Sciences, University of Toronto, Toronto, Canada, 3 Department of Psychiatry, University of Toronto, Toronto, Canada, 4 Banting and Best Diabetes Centre, University of Toronto, Toronto, Canada, 5 Human Biology Department, University of Toronto, Toronto, Canada, 6 Temerty Faculty of Medicine, University of Toronto, Toronto, Canada, 7 Brain Health Imaging Centre, Centre for Addiction and Mental Health (CAMH), Toronto, Canada, 8 Toronto Dementia Research Alliance, Temerty Faculty of Medicine, University of Toronto, Toronto, Canada

‡ SW and NS share first authorship on this work.
* mahavir.agarwal@camh.ca

**Data Availability Statement:** All relevant data are within the manuscript and its Supporting information files.

## Abstract

### Background

Aberrant brain insulin signaling has been posited to lie at the crossroads of several metabolic and cognitive disorders. Intranasal insulin (INI) is a non-invasive approach that allows investigation and modulation of insulin signaling in the brain while limiting peripheral side effects.

### Objectives

The objective of this systematic review and meta-analysis is to evaluate the effects of INI on cognition in diverse patient populations and healthy individuals.

### Methods

MEDLINE, EMBASE, PsycINFO, and Cochrane CENTRAL were systematically searched from 2000 to July 2021. Eligible studies were randomized controlled trials that studied the effects of INI on cognition. Two independent reviewers determined study eligibility and extracted relevant descriptive and outcome data.

### Results

Twenty-nine studies (pooled $N = 1,726$) in healthy individuals as well as those with Alzheimer's disease (AD)/mild cognitive impairment (MCI), mental health disorders, metabolic disorders, among others, were included in the quantitative meta-analysis. Patients with AD/MCI treated with INI were more likely to show an improvement in global cognition (SMD = 0.22, 95% CI: 0.05–0.38 $p$ = <0.00001, $N$ = 12 studies). Among studies with healthy

**Funding:** The authors received no specific funding for this work.

**Competing interests:** The authors have read the journal's policy and have the following competing interests: S.W. is supported by the Ontario Graduate Scholarship (OGS) and the Cleghorn Fellowship in Schizophrenia Studies outside of the submitted work. N.S. is supported by the Ontario Graduate Scholarship (OGS) and Banting and Best Diabetes Centre (BBDC) Novo-Nordisk Graduate Studentship outside of the submitted work. M.H. is supported in part by an Academic Scholars Award from the Department of Psychiatry, University of Toronto, and has grant support from the Banting and Best Diabetes Centre (BBDC), the Canadian Institutes of Health Research (CIHR), PSI Foundation, Ontario, holds the Kelly and Michael Meighen Chair in Psychosis Prevention, and the Cardy Schizophrenia Research Chair outside of the submitted work. She is also supported by the Danish Diabetes Academy, and a Steno Diabetes Centre Fellowship Award outside of the submitted work. T.K.R. has received research support from Brain Canada, Brain and Behavior Research Foundation, Bright Focus Foundation, Canada Foundation for Innovation, Canada Research Chair, Canadian Institutes of Health Research, Centre for Aging and Brain Health Innovation, National Institutes of Health, Ontario Ministry of Health and Long-Term Care, Ontario Ministry of Research and Innovation, and the Weston Brain Institute outside of the submitted work. G.R. has received research support from the Canadian Institutes of Health Research (CIHR), University of Toronto, Research Hospital Fund–Canada Foundation for Innovation (RHF-CFI), and HLS Therapeutics Inc. outside of the submitted work. S.M.A. is supported in part by an Academic Scholars Award from the Department of Psychiatry, University of Toronto, and has grant support from CIHR, PSI Foundation, Ontario, and the CAMH Discovery Fund outside of the submitted work. M.H. receives consultant fees from Alkermes Inc. G.R. has received advisory board support from HLS Therapeutics and consultant fees from Mitsubishi Tanabe Pharma Corporation outside of the submitted work. There are no patents, products in development or marketed products associated with this research to declare. This does not alter our adherence to PLOS ONE policies on sharing data and materials.

individuals and other patient populations, no significant effects of INI were found for global cognition.

## Conclusions

This review demonstrates that INI may be associated with pro-cognitive benefits for global cognition, specifically for individuals with AD/MCI. Further studies are required to better understand the neurobiological mechanisms and differences in etiology to dissect the intrinsic and extrinsic factors contributing to the treatment response of INI.

## 1. Introduction

For decades, the majority of insulin research has primarily focused on the action of insulin in peripheral tissues as the brain was long considered to be an insulin-insensitive organ. Insulin receptors are expressed widely throughout the brain, with notable concentrations in the olfactory bulb, cerebral cortex, striatum, hypothalamus, and hippocampus, thereby highlighting the role of insulin in processes such as cognition, appetite, and glucose regulation [1–4].

Central administration of insulin has been shown to play a prominent role in regulating cognition [5, 6]. This may be through insulin signalling pathways modulating processes of long-term potentiation and long-term depression or triggering release of various neurotrophic factors to promote neuronal survival [7, 8]. The role of CNS insulin on cognition has been further understood through several studies that suggest attenuated insulin action in the brain may be a critical factor in the development of age-related cognitive decline and Alzheimer's disease (AD). To this point, the central insulin resistance hypothesis asserts that lack of insulin responsivity in the brain may be implicated in the pathogenesis of various clinical conditions including Parkinson's disease, schizophrenia, dementia, depression, and type 2 diabetes mellitus (T2DM) [9–13]. As such, aberrant brain insulin signaling has been posited to lie at the crossroads of metabolic and cognitive disorders.

As impaired CNS insulin function has been observed in many pathologies, greater focus has been placed on using central insulin as a potential treatment intervention. Intranasal insulin (INI) is a non-invasive technique that delivers insulin directly to the CNS while limiting peripheral spill over [14]. A recent systematic review and meta-analysis sought to evaluate the effectiveness of INI on cognition in patients with mild cognitive impairment (MCI) or dementia; notably, results did not reveal a significant effect of INI on cognitive functioning compared to placebo in these populations [15]. However, the effect of INI on cognition across all population groups over the last two decades remains to be systematically reviewed. The objective of this systematic review and meta-analysis is to synthesize available evidence evaluating the effects of INI on cognition in diverse patient populations and cognitively unimpaired, healthy individuals.

## 2. Materials and methods

We conducted a systematic review and meta-analysis in accordance with the Preferred Reporting Items for Systematic Reviews and Meta-Analyses (PRISMA) methodology and reporting standard. A review protocol was submitted to the PROSPERO international database of prospectively registered systematic on July 22, 2021 (PROSPERO #CRD42021262478). Our original search was conducted on July 22, 2021, and an updated search was completed on February 11, 2023.

## 2.1. Search strategy

MEDLINE, Embase, PsycINFO, CINAHL, CENTRAL, ClinicalTrials.gov, and the ICTRP Search Portal were systematically searched for relevant peer-reviewed studies to be included in the review (S1 Table in S1 File).

## 2.2. Study selection

Three authors (NS, SW, ZE) independently screened all identified articles from the systematic search; agreement from a minimum of two authors was required for a study to be either included or excluded. Any disagreements were resolved by re-checking source papers and discussion between all authors. Studies were selected according to the following inclusion criteria:

1. *Study Design*: Any randomized controlled trial (RCT) that included a placebo control group. We excluded all other study types, including cross-sectional studies, observational studies, case reports, opinions, commentaries, editorials, replies, letters to the editor, and incomplete studies.

2. *Study Population*: Populations of interest included, but were not limited to, healthy controls, mental health disorders, metabolic disorders, neurodegenerative and neurodevelopmental disorders. There were no restrictions based on age or sex.

3. *Intervention*: Studies could examine the effects of regular human insulin or analog insulin lispro, delivered intranasally, on various domains of cognition. Once again, studies must have included a placebo control group, while any study examining other intranasal peptide interventions were excluded.

4. *Outcomes*: The studies were required to evaluate the effects of INI administration on standardized cognitive outcomes of interest.

## 2.3. Data extraction

All data were independently extracted and reviewed by three authors (NS, SW, ZE). Corresponding authors were contacted if data could not be extracted in a usable form from the published paper.

## 2.4. Bias assessment

The Cochrane Risk of Bias (RoB) tool was used to assess bias in the context of our outcomes of interest (i.e., cognition) [16]. Seven evidence-based domains were assessed, including random sequence generation, allocation concealment, blinding of participants and personnel, blinding of outcome assessment, incomplete outcome data, selective reporting, and other bias. RoB assessment was conducted by three independent reviewers (NS, SW, JN), with two individuals assigned per study; conflicts were resolved through group discussion and consensus. Sensitivity analyses were conducted to exclude lower quality studies in which three or more of the domains were labelled as 'high' or 'unclear' risk. A funnel plot of the included studies was used to assess for publication bias.

## 2.5. Outcome measures

The primary outcome was the effect of INI on cognition as assessed by individual cognitive domains including working memory, verbal working memory, verbal memory, verbal fluency, visual working memory, visual learning and memory, declarative memory, nondeclarative memory, hippocampal-dependent memory, executive function, attention, inhibitory control,

and dementia (S3 Table in S1 File). A measure of global cognition was calculated by averaging the effect sizes of all individual cognitive domains for each study [17]. The effect size for global cognition for each study was then entered into another meta-analysis to obtain a pooled effect of global cognition across studies for each patient population. Note, the number of individual cognitive domains varied per study. See S3 Table in S1 File for all the neuropsychological tests used to measure these cognitive domains. Secondary outcome measures included serum insulin and glucose levels and side effects of INI, as available. Subgroup analyses for age, sex, and dose of INI were also conducted, as available, to explore potential sources of heterogeneity.

## 2.6. Synthesis of results

Quantitative data from all studies were pooled in a random effects meta-analysis using Review Manager 5.4. A minimum of two studies was required to meta-analyze effects; single study comparisons were reported narratively. Standardized mean difference (SMD) was calculated when pooling studies together with different outcome measures for a specific cognitive domain, while mean difference (MD) was calculated for studies using the same outcome measure. MD was calculated for physiological outcomes, including serum insulin and glucose levels following INI administration. Odds ratios (OR) were used for dichotomous data (e.g., side effects). Endpoint and change score data were combined in the analysis. Separate comparisons of INI versus placebo on each cognitive outcome measure were conducted for each clinical population and healthy individuals. Given the limited number of studies with available data, comparisons of adverse effects and physiological blood values were conducted across patient populations. Heterogeneity was assessed using the $I^2$ statistic [18]. Publication bias for comparisons with 10 or more included studies was assessed using funnel plots.

## 3. Results

Our initial search identified 2654 results. Following title and abstract screening, 52 studies were assessed for full-text eligibility. A total of 32 studies met inclusion criteria for this review and 26 of these studies, with a total of 1,726 patients and healthy individuals, were included in the quantitative meta-analysis (Fig 1; PRISMA flowchart).

### 3.1. Study characteristics

The 32 RCTs included in this review were published between 2004 and 2021. A variety of populations were represented, including: healthy individuals ($N$ = 11) [19–29], MCI and AD ($N$ = 12) [30–42], schizophrenia ($N$ = 2) [43, 44], bipolar disorder ($N$ = 1) [45], major depressive disorder ($N$ = 1) [46], obese men ($N$ = 1) [47], Down syndrome ($N$ = 1) [48], type 2 diabetes ($N$ = 2) [26, 28], Phelan-McDermid syndrome ($N$ = 1) [49], and Parkinson's disease ($N$ = 1) [50]. The median dose of INI was 40 IU (range 40 to 160 IU). Ten studies used an acute (single dose) of INI, while the remaining studies administered INI over a longer term. The median duration of INI treatment in the long-term studies was 8 weeks (range 1 to 52 weeks). The mean age of all populations included in our review was 53.4 years. Characteristics of all included studies can be found in S2 Table in S1 File.

### 3.2. Meta-analysis of global cognition by patient population

**3.2.1. Healthy individuals.** Global cognition for healthy, cognitively unimpaired individuals were pooled ($k$ = 11; $N$ = 400) [19–29]. No significant effects of INI were found for global cognition (SMD 0.02, 95% CI -0.05 to 0.09; $I^2$ = 45%) (Fig 2). Among the healthy control studies, five studies examined the effects of intranasal regular human insulin versus placebo [19,

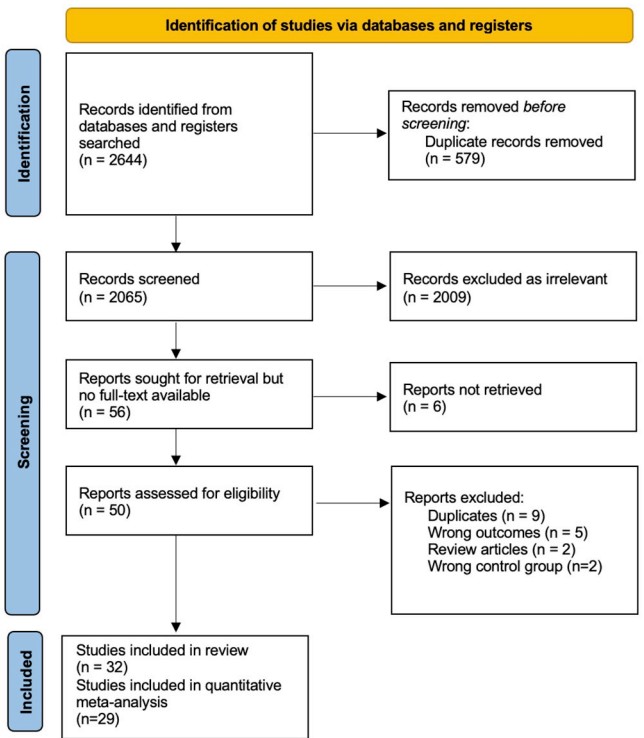

**Fig 1. PRISMA flow chart.** Preferred Reporting Items for Systematic Reviews and Meta-Analyses (PRISMA) flow chart of included studies.

20, 24, 25, 28], one study examined both insulin aspart and regular human insulin versus placebo [21], two studies examined the effects of insulin actrapid versus placebo [27, 29], while another study examined 20, 40, 60, 80, 100, and 120 IU of insulin aspart versus placebo; each dose group is reported separately [22]. Seven studies administered an acute dose of INI [19, 22–27], and four studies administered INI over a longer period of treatment [19, 20, 28, 29]. There were no significant differences noted between studies that used an acute dose versus long-term dose.

**3.2.2. All patient populations.** Patient populations were categorized into four groups: mental health disorders, AD and MCI, metabolic disorders, and other disorders (Fig 3). These are reported independently in the analysis. Treatment duration ranged from acute (single dose) administration to 4 months. Four studies examined the effects of INI on cognition in patients with mental health disorders including schizophrenia [43, 44], bipolar disorder [45], and major depressive disorder [46]. There were no significant effects observed for global cognition following INI treatment in patients with mental health disorders (SMD 0.07, 95% CI -0.09 to 0.24; $I^2$ = 0%) (Fig 3). Eleven studies examined the effects of INI on global cognition outcomes in patients with AD/MCI using the Alzheimer's Disease Assessment Scale-Cognitive Subscale (ADAS-Cog) [30–42]. Pooling these studies together revealed a significant improvement in global cognition following treatment with INI (SMD 0.22, 95% CI 0.05 to 0.38; $I^2$ = 69%) (Fig 3). Three studies reported on the effects of INI on cognition in patients with metabolic disorders, one in obese men and two in adults with T2DM [26, 28, 47]. No significant effect was observed for global cognition in patients with metabolic disorders (SMD 0.18, 95% CI -0.11 to 0.48; $I^2$ = 0%) (Fig 3). Global cognition for two studies were pooled together under

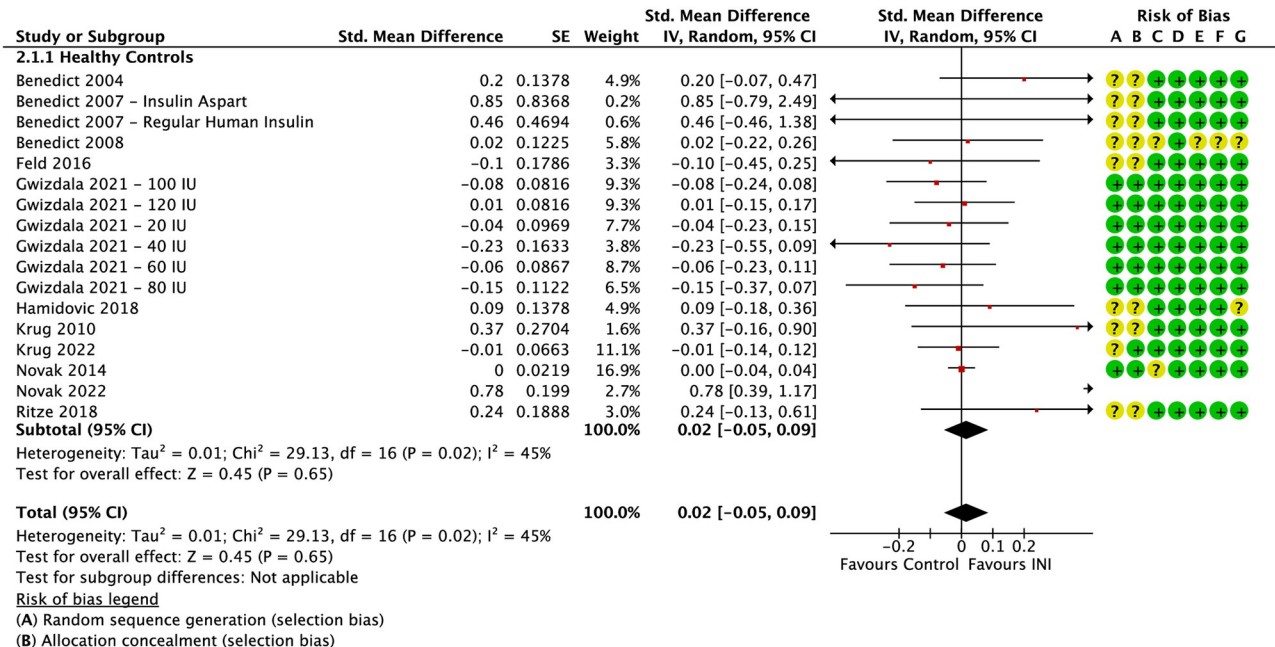

**Fig 2. Effect of INI vs. placebo on global cognition in healthy individuals.** Forest of plot and risk of bias assessment for the effect of INI vs. placebo on global cognition in healthy individuals.

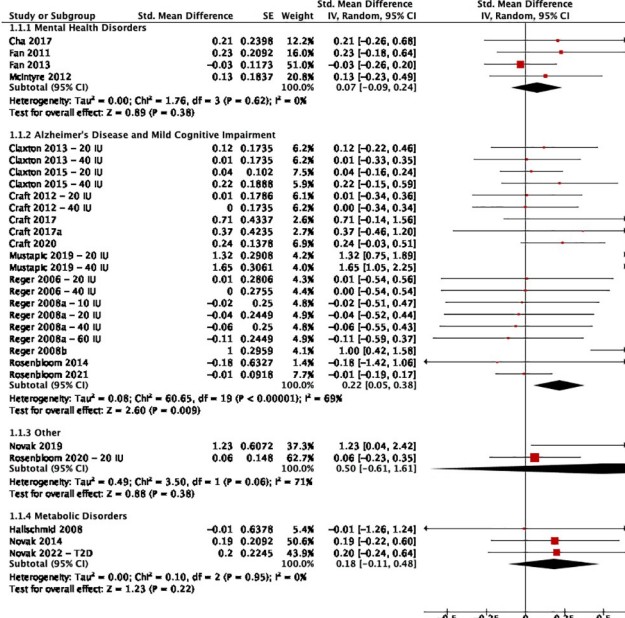

**Fig 3. Effect of INI vs. placebo on global cognition in all patient populations.** Forest plot and risk of bias assessment for the effect of INI vs. placebo on global cognition in all patient populations.

other disorders. One study explored the effects of 8 weeks of 20 IU of INI on cognition in individuals with Down syndrome, and the other study explored the acute effects of 40 IU of INI in patients with Parkinson's disease [26, 48]. INI did not improve global cognition in either of these patient populations (SMD 0.50, 95% CI -0.61 to 1.61; $I^2$ = 71%).

To further investigate whether INI improved a specific cognitive outcome for patients with AD/MCI, the studies were pooled together to evaluate the effect of INI on the following cognitive outcome measures: verbal working memory, attention, verbal memory, executive function, visuospatial memory, and global cognition. However, there were no significant effects observed for any of these outcomes.

### 3.3. Meta-analysis of INI related side effects (all populations)

Adverse events following INI administration were reported in 22 of the included studies. Common side effects included nasal irritation/rhinitis, light-headedness/dizziness, nausea, and nose bleeds (S2 Table in S1 File). There was no significant difference in number of reports for any of these side effects between the INI and placebo groups (S1-S4 Figs in S1 File). There was also no significant difference between groups in terms of total number of adverse events being reported (S5 Fig in S1 File).

### 3.4. Meta-analysis of post-INI serum insulin and glucose levels (all populations)

Serum concentrations of glucose and insulin levels following INI administration were reported in 20 of the included studies, 9 of which provided data to pool in a meta-analysis for each outcome (S2 Table in S1 File). Across all populations, serum insulin and glucose levels were not significantly affected following INI administration, as compared to placebo (S6, S7 Figs in S1 File).

### 3.5. Subgroup analysis of age, sex, and dose of INI

Within healthy individuals, age was not a heterogenous variable (S8 Fig in S1 File). Given our study populations, subgroup analysis could not be conducted for patients with AD/MCI as they are all classified as older adults (≥65 years old). Across studies with healthy individuals, INI improved global cognition in studies comprised of >50% males (S9 Fig in S1 File). In contrast, INI was found to improve global cognition in patients with AD/MCI in studies comprised of >50% males (S10 Fig in S1 File). Moreover, 40 IU of INI was shown to improve global cognition in patients with AD/MCI compared to other administered doses (10 IU, 20 IU, and 60 IU) (S11 Fig in S1 File). Subgroup analysis could not be conducted for the mental health and metabolic disorder groups due to the limited number of studies within each category. Across all patient populations, 4 studies used administered an acute dose of INI, while the remaining 15 had a longer intervention period (range 3 weeks to 4 months). Significant effects on global cognition were only noted in the long-term administration studies (SMD 0.26, 95% CI 0.10 to 0.42, $I^2$ = 70%) in comparison to the acute studies (SMD 0.02, 95% CI -0.13, 0.17, $I^2$ = 0%) (S12 Fig in S1 File).

### 3.6. Qualitative reporting of outcomes

A total of 3 studies retrieved from the search did not report data in a way that could be extracted and synthesized in a meta-analysis [38, 42, 49]. Two out of the three studies examined the effects of prolonged INI treatment on cognition in patient populations with AD and MCI [38, 42], while the remaining study assessed this in children with Phelan McDermid

syndrome [49]. Both AD/MCI studies reported improvements in ADAS-Cog scores with long-term INI treatment compared to placebo. However, one of these studies observed that INI did not result in meaningful improvements in immediate and delayed recall in mild-moderate AD patients [42]. In the third study, INI was observed to improve the cognition and social skills of children with Phelan McDermid syndrome over the age of 3 years old [49]. S2 Table in S1 File summarizes the characteristics of the studies included in this section, along with their main findings.

### 3.7. Risk of bias assessment

There were no studies included in the quantitative meta-analysis that were deemed to be of low quality or have a high risk of bias (S13 Fig in S1 File). As such, there was no need to conduct any sensitivity analyses for any of the reported outcome effects. Three studies that were only reported narratively [34, 36, 41] had three categories with "unclear risk" due to lack of information in their published articles, and one category deemed "high risk". Publication bias could not be assessed using funnel plots given the limited number of studies included within each comparison. No substantial publication bias was detected in these comparisons (S14 and 15 Figs in S1 File).

## 4. Discussion

In this review, we present a comprehensive overview and analysis of RCTs evaluating the effects of INI administration on various domains of cognition across all ages and populations. We demonstrate that INI may be associated with pro-cognitive benefits for pooled global cognition in patients with AD and MCI, while there was no significant effect observed for individual cognitive subdomains. Additionally, long-term administration of INI may provide more therapeutic benefits than acute administration. To the best of our knowledge, this is the first review to pool together data in healthy individuals and patient populations, thus providing meaningful preliminary evidence for this novel and emerging field. We have also identified several gaps and unanswered questions that should be addressed in future research to propel this field forward.

The cognitive benefits of INI were only found in certain populations in our study, specifically patients with AD and MCI. Interestingly, this finding is in contrast to a previous meta-analysis in the AD/MCI population that failed to detect a significant difference on global cognition with INI versus placebo [15]. However, it is possible that this finding may be due to the inclusion of two studies in that review for which the primary diagnosis was not AD/MCI [46, 51]. In this review, Cha et al. (2017) was included in the comparison for mental health disorders [46], and this study showed no effect of INI therapy on any of the cognitive outcomes for patients with MDD. The second study was an unpublished report of patients with HIV-associated neurocognitive disorder [51], which also did not demonstrate an effect of INI on cognition [15]. Taken together, the pro-cognitive effect of INI in AD/MCI is presently inconclusive, and further research is warranted to elucidate the therapeutic effects of INI.

In exploring the effects of INI on cognition in AD/MCI, there are important moderators to consider. For example, the apolipoprotein E4 (APOE-ε4) carrier status of an individual may introduce differences in insulin sensitivity between APOE-ε4 carriers versus non-carriers. In fact, the occurrence of insulin resistance is greatest in adults with AD who are APOE-ε4 non-carriers [52]. In the present review, there were four AD/MCI studies that stratified patients according to their APOE-ε4 carrier status [33, 36, 41, 53], with the remaining AD studies not indicating the APOE-ε4 status of the sample. In three of these studies, INI acutely facilitated memory improvements in APOE-ε4− memory-impaired subjects, as compared to APOE-ε4

+ subjects who showed no benefits or a decline in memory [34, 36, 41]. The last study failed to note a main effect of APOE-ε4 status on therapeutic response [33]. This demonstrates that there may be identifiable heterogeneity within specific populations and that INI may only be beneficial for a subgroup of patients. Thus, stratifying patients according to genotype may help identify individuals that would be most responsive to this intervention. More research is required to understand the mechanisms through which the APOE genotype attenuates the cognitive effects of INI [41].

There were no significant effects of INI observed in any of the other comparisons included in the present study. This may be attributed to illness or treatment-specific characteristics within each population that may moderate the response to INI and limit its generalizability across patient populations. One point to consider is the baseline level of cognitive functioning in a given population. For example, there may be varying levels of baseline neurocognitive deficits across the patient populations included in the mental health disorders comparison [54, 55]. McIntyre et al. (2012) demonstrated an improvement in executive functioning (Trail Making Test Part-B) with an acute dose of INI in patients with bipolar I/II disorder [45], whereas Fan et al. (2011 and 2013) did not observe any pro-cognitive improvements with INI in patients with schizophrenia [43, 44]. This might suggest that patients with higher baseline cognitive functioning may be more likely to benefit from INI in comparison to patients exhibiting more severe neurocognitive impairment.

In general, INI was well tolerated in both healthy individuals and across clinical populations. There were no significant differences in the number of side effects reported between the INI treatment and placebo groups, demonstrating the safety and tolerability of INI. In the present review, only 18 out of the 29 RCTs measured serum concentrations of glucose and insulin following INI administration. Among the seven studies that were pooled across all study populations, serum insulin and glucose levels were not significantly affected after INI administration. It should be noted that dose-dependent peripheral spillover has been reported in the literature. Higher doses of INI may temporarily increase circulating insulin and decrease blood glucose levels increasing the risk of hypoglycemia [56, 57]. While these reports of peripheral spillover did not cause any adverse side effects, these results highlight the importance of collecting serum glucose, insulin, and c-peptide concentrations after intranasal administration to ensure the safety of its use. Furthermore, our subgroup analyses demonstrated that sex and dose may affect the outcomes of INI on cognition. Thus, future studies should stratify their study population by sex to examine sex-differences as well the dose required for cognitive response to INI treatment.

There are some limitations to this study which must be addressed. Our results revealed moderate to substantial heterogeneity for specific study populations including the healthy controls ($I^2$ = 45%), AD/MCI ($I^2$ = 69%), and other disorders ($I^2$ = 71%). This may be attributed to the high heterogeneity in cognitive measures employed to assess the various domains of cognition. The lack of standardized reporting for cognitive outcomes made it difficult to compare the effects of INI on cognition between healthy individuals and across patient populations, precluding conclusions from being made. Furthermore, the lack of standardized reporting in the literature precluded data from being extracted from many of the identified studies, which limits the power of our results. Similarly, there were a small number of studies that examined a particular cognitive subdomain, which further limited the power of our analyses for individual cognitive outcomes. Thus, we pooled together different cognitive subdomains to obtain global cognition for each study. Second, there was a lack of stratification between sex and genotype in most studies, which may contribute as a potential source of heterogeneity and represent as significant confounders of cognitive outcomes as discussed above. Differential cognitive response to INI treatment may result from the observable

heterogeneity in experimental design across studies, such as type of insulin administered, treatment duration, and dose. Furthermore, our systematic review consists of a heterogenous sample spanning cognitive unimpaired individuals to multiple patient populations; thus, the pro-cognitive benefits of INI in one population may not be generalizable to other populations. To note, our meta-analysis categorized patients with Down syndrome and Parkinson's disease under other disorders. Therefore, the substantial heterogeneity observed may be attributed to the different patient population as well as the type of cognitive impairment. Third, not all included studies reported on BMI of the sample, and among those that did, there were variable effects observed of INI on cognitive outcomes. It is well established in the literature that obesity not only induces peripheral insulin resistance but can also induce insulin resistance within the brain [58]. Therefore, baseline BMI may play a role in mediating cognitive outcomes and influence INI treatment response. Future studies should include an assessment to better understand the metabolic profile of patients or healthy individuals to further explore the relationship between obesity, insulin resistance, and cognitive decline. Results from a systematic review based on studies with great diversity should be interpreted cautiously as it can bring a higher risk for extrapolation. Further research is warranted to better understand the neurobiological mechanisms and differences in etiology to dissect the intrinsic and extrinsic factors contributing to the treatment response of INI. We encourage investigators of upcoming RCTs to combine neuroimaging approaches with cognitive assessments to identify the neural correlates associated with INI therapy.

## 5. Conclusion

This systematic review and meta-analysis demonstrate that INI may be associated with pro-cognitive benefits, specifically for pooled global cognition. However, this effect is limited to patients with AD/MCI. Research over the last two decades has identified the CNS as an emerging major contributing site of insulin action. Taken together, and in keeping with recent clinical trials, our findings demonstrate that INI can be safely tolerated, and has the potential to improve memory by directly reaching brain regions involved in the regulation of cognition. As this is still a novel field of study, more research is required to understand the heterogeneity in treatment response of INI to extend the pro-cognitive benefits across different patient populations with the ultimate goal to improve their overall quality of life.

## Supporting information

**S1 File.**
(PDF)

**S1 Checklist. PRISMA 2020 checklist.**
(DOCX)

## Author Contributions

**Conceptualization:** Sally Wu, Nicolette Stogios, Margaret Hahn, Sri Mahavir Agarwal.

**Data curation:** Sally Wu, Nicolette Stogios, Janani Navagnanavel, Zahra Emami.

**Formal analysis:** Sally Wu, Nicolette Stogios.

**Investigation:** Sally Wu, Nicolette Stogios, Sri Mahavir Agarwal.

**Methodology:** Sally Wu, Nicolette Stogios, Margaret Hahn, Sri Mahavir Agarwal.

**Software:** Sally Wu, Nicolette Stogios.

**Supervision:** Margaret Hahn, Sri Mahavir Agarwal.

**Validation:** Sally Wu, Nicolette Stogios, Janani Navagnanavel, Zahra Emami.

**Visualization:** Sri Mahavir Agarwal.

**Writing – original draft:** Sally Wu, Nicolette Stogios, Janani Navagnanavel, Zahra Emami.

**Writing – review & editing:** Sally Wu, Nicolette Stogios, Margaret Hahn, Araba Chintoh, Philip Gerretsen, Ariel Graff-Guerrero, Tarek K. Rajji, Gary Remington, Sri Mahavir Agarwal.

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
