## [Decision Letter · Decision Letter 0]

16 Apr 2023

PONE-D-23-05529Outcomes and Clinical Implications of Intranasal Insulin on Cognition in Humans:

A Systematic Review and Meta-AnalysisPLOS ONE

Dear Dr. Agarwal,

Thank you for submitting your manuscript to PLOS ONE. After careful consideration, we feel that it has merit but does not fully meet PLOS ONE’s publication criteria as it currently stands. Therefore, we invite you to submit a revised version of the manuscript that addresses the points raised during the review process.

We look forward to receiving your revised manuscript.

Kind regards,

Belgin Sever, Ph.D.

Academic Editor

PLOS ONE

Reviewers' comments:

Reviewer's Responses to Questions

**Comments to the Author**

1. Is the manuscript technically sound, and do the data support the conclusions?

Reviewer #1: Yes

Reviewer #2: No

2. Has the statistical analysis been performed appropriately and rigorously? 

Reviewer #1: Yes

Reviewer #2: No

3. Have the authors made all data underlying the findings in their manuscript fully available?

Reviewer #1: Yes

Reviewer #2: No

4. Is the manuscript presented in an intelligible fashion and written in standard English?

Reviewer #1: Yes

Reviewer #2: No

5. Review Comments to the Author

Reviewer #1: Would like initially to congratulate the authors on performing hard work in preparing this manuscript after their review and analysis of many of the screened studies. This would be considered a well done manuscript for future references.

Reviewer #2: Reviewer statement: The manuscript entitled “Outcomes and Clinical Implications of Intranasal Insulin on Cognition in Humans: A systematic review and metanalysis” aims the application of metanalysis technics to test the association of intranasal insulation and cognition. The manuscript carries some methodological flaws that need to be addressed before being accepted for publication.

Main suggestions:

1- The version I received had the actual manuscript text duplicated. Something went wrong during the manuscript final editing.

2- The grammar and phrase constructions used in the manuscripts need to be revised.

3- The definition of the global cognition phenotype is not clear. The authors pooled data from different studies, each one with their own unique set of measured cognitive traits. The authors need to show the set of cognitive traits shared among the individual studies and how those study-specific traits were meta-analyzed.

4- The authors never considered the effect of simple covariates such as sex and age on each individual study. The individual studies are focused on different phenotypes with completely different cohorts. This heterogeneity and not considering the effect of sex and age could be a major source of bias that would be even more pronounced when doing a metanalysis.

5- The authors merged studies focused on mental health disorders, AD and metabolic disorders (lines 177 to 196) without considering that the underlying conditions may have an effect on the “global cognition”. This lack of rigor can lead to many experimental artifacts.

6- The authors didn’t explain the assumptions if the Cochrane Risk of Bias tool and didn’t even include a scientific reference for it.

7- The authors never even discussed the heterogeneity statistics I2 on each individual metanalysis. Low I2 statistics hinders the interpretation of any metanalysis.

8- There is a lot of confusion between the terms metanalysis and pooled analysis during the reading. Terms such as pooled population metanalysis are very misleading.

9- The authors need to further develop their introduction and their discussion sections. They need to incorporate similar studies using the same metanalytic approaches even for other focal phenotypes.

10- There is a lot of confusion between the definition of tables and a figure. Figure 2 and 3, as an example, are tables and not figures. The authors need to split tables and figures and discussed The authors didn’t include labels to any “figure” or supplementary “figures”. The interpretation is challenging without the proper labels.

11- The authors referred (line 236) the supplementary figure 9 that didn’t exist in the supplementary materials.

6. PLOS authors have the option to publish the peer review history of their article (what does this mean?). If published, this will include your full peer review and any attached files.

Reviewer #1: No

Reviewer #2: No

---

## [Author Response · Author response to Decision Letter 0]

11 May 2023

The version I received had the actual manuscript text duplicated. Something went wrong during the manuscript final editing.

There are two versions of the manuscript. The latter version has tracked changes with our updated systematic search performed on February 11, 2023, as requested by the journal. The manuscript and analyses were then updated accordingly. Per journal protocols, a tracked and clean version of the manuscript were uploaded upon resubmission. 

The grammar and phrase constructions used in the manuscripts need to be revised.

Thank you for your comment. We have gone through the manuscript and proofread to ensure that there are no grammatical and phrase construction errors. 

The definition of the global cognition phenotype is not clear. The authors pooled data from different studies, each one with their own unique set of measured cognitive traits. The authors need to show the set of cognitive traits shared among the individual studies and how those study-specific traits were meta-analyzed.

Supplementary Table 3 outlines the different cognitive domains that were measured across studies as well as the various cognitive measures used to assess each domain. 

For each included study in the quantitative meta-analysis, an effect size for global cognition was calculated by averaging the effect size of individual cognitive domains. The effect size for global cognition for each study was then entered into another meta-analysis to obtain a pooled effect of global cognition across all studies in each patient population. We have updated and included this explanation in our Methods section to further elaborate our definition of global cognition. 

The authors never considered the effect of simple covariates such as sex and age on each individual study. The individual studies are focused on different phenotypes with completely different cohorts. This heterogeneity and not considering the effect of sex and age could be a major source of bias that would be even more pronounced when doing a metanalysis.

Based on the reviewer’s comment, we ran subgroup analyses based on sex, age, and dose of INI. Within patient populations, age wasn’t a heterogenous variable. Subgroup analysis could be conducted for patients with AD/MCI as all patients were 55 years or older. Furthermore, subgroup analysis could not be conducted due to the limited number in the mental health and metabolic disorder groups. Please see section 3.5 for the updated results.

The authors merged studies focused on mental health disorders, AD and metabolic disorders (lines 177 to 196) without considering that the underlying conditions may have an effect on the “global cognition”. This lack of rigor can lead to many experimental artifacts.

Thank you for your comment. We agree with the reviewer that this is a heterogenous population and removed the overall pooled effect size in Figure 3. We reported on the outcomes individually, though they are displayed in one figure for organization purposes. 

The authors didn’t explain the assumptions if the Cochrane Risk of Bias tool and didn’t even include a scientific reference for it.

Thank you for bringing this to our attention. We have added the reference for the Cochrane Risk of Bias tool as well as the different domains it addresses. 

The authors never even discussed the heterogeneity statistics I2 on each individual metanalysis. Low I2 statistics hinders the interpretation of any metanalysis.

Thank you for your comment. We have addressed the heterogeneity I2 statistics in our limitation sections. We also discuss potential sources of heterogeneity throughout our discussion based on each patient population. 

There is a lot of confusion between the terms metanalysis and pooled analysis during the reading. Terms such as pooled population metanalysis are very misleading.

Thank you for bringing this to our attention. We have updated our results section using the proper nomenclature for a meta-analysis. 

The authors need to further develop their introduction and their discussion sections. They need to incorporate similar studies using the same metanalytic approaches even for other focal phenotypes.

Thank you for your suggestion. We have further elaborated on the study that we referenced for the same meta-analytic approach.

There is a lot of confusion between the definition of tables and a figure. Figure 2 and 3, as an example, are tables and not figures. The authors need to split tables and figures and discussed The authors didn’t include labels to any “figure” or supplementary “figures”. The interpretation is challenging without the proper labels.

As per requested by PLOS One, the figure captions are included in the manuscript and the figures followed the references at the end. To note, Figure 2 and 3 are forest plots that were exported by RevMan. Forest plots are a graphical representation of the findings of multiple studies, thus we listed them as figures and not tables. All figures and tables are correctly labeled in our supplementary file. 

The authors referred (line 236) the supplementary figure 9 that didn’t exist in the supplementary materials.

Thank you for bringing this to our attention. We had mislabeled the last few figures. We have updated the supplementary file with additional subgroup analyses and updated the figure label accordingly.

---

## [Editor Report · Decision Letter 1]

25 May 2023

Outcomes and Clinical Implications of Intranasal Insulin on Cognition in Humans: A Systematic Review and Meta-Analysis

PONE-D-23-05529R1

Dear Dr. Agarwal,

We’re pleased to inform you that your manuscript has been judged scientifically suitable for publication and will be formally accepted for publication once it meets all outstanding technical requirements.

Kind regards,

Belgin Sever, Ph.D.

Academic Editor

PLOS ONE
---

## [Editor Report · Acceptance letter]

2 Jun 2023

PONE-D-23-05529R1 

Outcomes and Clinical Implications of Intranasal Insulin on Cognition in Humans: A Systematic Review and Meta-Analysis 

Dear Dr. Agarwal:

I'm pleased to inform you that your manuscript has been deemed suitable for publication in PLOS ONE. Congratulations! Your manuscript is now with our production department. 

Kind regards, 

on behalf of

Assoc. Prof. Dr. Belgin Sever 

Academic Editor

PLOS ONE